# Visual Imitation Learning with Patch Rewards

**Minghuan Liu**[1,2,†]**, Tairan He**[1]**, Weinan Zhang**[1,✉]**, Shuicheng Yan**[2]**, Zhongwen Xu**[2]
[1]Shanghai Jiaotong University, [2]Sea AI Lab
{minghuanliu, whynot, wnzhang}@sjtu.edu.cn, {yansc,xuzw}@sea.com

## Abstract

Visual imitation learning enables reinforcement learning agents to learn to behave from expert visual demonstrations such as videos or image sequences, without explicit, well-defined rewards. Previous research either adopted supervised learning techniques or induce simple and coarse scalar rewards from pixels, neglecting the dense information contained in the image demonstrations. In this work, we propose to measure the expertise of various local regions of image samples, or called *patches*, and recover multi-dimensional *patch rewards* accordingly. Patch reward is a more precise rewarding characterization that serves as a fine-grained expertise measurement and visual explainability tool. Specifically, we present Adversarial Imitation Learning with Patch Rewards (PatchAIL), which employs a patch-based discriminator to measure the expertise of different local parts from given images and provide patch rewards. The patch-based knowledge is also used to regularize the aggregated reward and stabilize the training. We evaluate our method on DeepMind Control Suite and Atari tasks. The experiment results have demonstrated that PatchAIL outperforms baseline methods and provides valuable interpretations for visual demonstrations. Our codes are available at https://github.com/sail-sg/PatchAIL.

## 1 Introduction

Reinforcement learning (RL) has gained encouraging success in various domains, *e.g.*, games (Silver et al., 2017; Yang et al., 2022; Vinyals et al., 2019), robotics (Gu et al., 2017) and autonomous driving (Pomerleau, 1991; Zhou et al., 2020), which however heavily relies on well-designed reward functions. Beyond hand-crafting reward functions that are non-trivial, imitation learning (IL) offers a data-driven way to learn behaviors and recover informative rewards from expert demonstrations without access to any explicit reward (Arora & Doshi, 2021; Hussein et al., 2017). Recently, visual imitation learning (VIL) (Pathak et al., 2018; Rafailov et al., 2021) has attracted increasing attention, which aims to learn from high-dimensional visual demonstrations like image sequences or videos. Compared with previous IL works tackling low-dimensional inputs, *i.e.* proprioceptive features like positions, velocities, and accelerations (Ho & Ermon, 2016; Fu et al., 2017; Liu et al., 2021), VIL is a more general problem in the physical world, such as learning to cook by watching videos. For humans, it is very easy but it remains a critical challenge for AI agents.

Previous approaches apply supervised learning (*i.e.* behavior cloning (Pomerleau, 1991; Pathak et al., 2018)) or design reward functions (Reddy et al., 2019; Brantley et al., 2019) to solve VIL problems. Among them, supervised methods often require many expert demonstrations and suffer from compounding error (Ross et al., 2011), while the rest tend to design rewards that are too naive to provide appropriate expertise measurements for efficient learning. Some recent works (Rafailov et al., 2021; Cohen et al., 2021; Haldar et al., 2022) choose to directly build on pixel-based RL methods, *i.e.* using an encoder, and learn rewards from the latent space. These methods inspect the whole image for deriving the rewards, which tend to give only coarse expertise measurements. Additionally, they focus on behavior learning, and seldom pay any attention to the explainability of the learned behaviors.

In this work, we argue that the scalar-form reward used in previous works only conveys "sparse" information of the full image, and hardly measures local expertise, which limits the learning effi-

---

†The work was done during an internship at Sea AI Lab, Singapore. ✉Corresponding author.

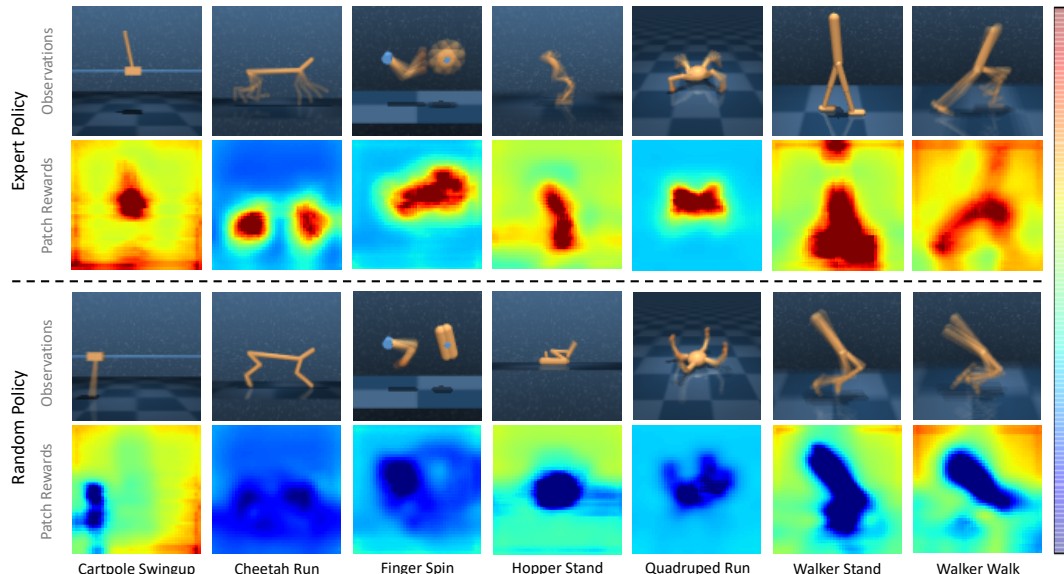

Figure 1: Visualization of patch rewards by the proposed PatchAIL on seven DeepMind Control Suite tasks. The patch rewards are calculated regarding several stacked frames and are mapped back on each pixel weighted by the spatial attention map. We use the final model trained with around 200M frames. Strikingly, the patch rewards align with the movement of key elements (*e.g.*, moving foot, inclined body) on both expert and random samples, indicating where the high rewards (red areas) or low ones (blue areas) come from, compared with the reward on backgrounds. With the patch rewards recovered from expert's visual demonstrations, we can easily reason why and where agents are rewarded or penalized in these tasks compared with expert demonstrations without any task prior. Figure best viewed in color. A corresponding video is provided in the demo page https://sites.google.com/view/patchail/.

ciency. Instead of inducing a scalar reward from the whole image, our intuition is to measure the expertise of different local regions of the given images, or what we call *patches*, and then recover the multi-dimensional rewards, called *patch rewards* accordingly. Compared with scalar rewards, patch rewards provide a more precise rewarding characterization of visual control tasks, thus supporting efficient IL training and visual explainability of the learned behaviors.

We then propose to achieve more efficient VIL via inferring dense and informative reward signals from images, which also brings better explainability of the learned behaviors. To achieve this goal, we develop an intuitive and principled approach for VIL, termed Adversarial Imitation Learning with Patch rewards (PatchAIL), which produces patch rewards for agents to learn by comparing local regions of agent image samples with visual demonstrations, and also provides interpretability of learned behaviors. Also, it utilizes a patch regularization mechanism to employ the patch information and fully stabilize the training. We visualize the patch rewards induced by our PatchAIL in Fig. 1. From the illustration, we can easily interpret where and why agents are rewarded or penalized in these control tasks compared with expert demonstrations without any task prior. These examples show the friendly explainability of our method, and intuitively tell why the algorithm can work well.

We conduct experiments on a wide range of DeepMind Control Suite (DMC) (Tassa et al., 2018) tasks along with several Atari tasks, and find our PatchAIL significantly outperforms all the baseline methods, verifying the merit of our PatchAIL for enhancing the learning efficiency of VIL. Moreover, we conduct ablations on the pivotal choices of PatchAIL, showing the key options in PatchAIL such as the number of patches and aggregation function highly impact the learning performance. To verify its explainability, we further compare the spatial attention maps of PatchAIL to baseline methods, showing PatchAIL is better at focusing on the correct parts of robot bodies. It is hoped that our results could provide some insights for exploiting the rich information in visual demonstrations to benefit VIL and also reveal the potential of using patch-level rewards.

## 2 PRELIMINARIES

**Markov decision process.**   RL problems are typically modeled as a $\gamma$-discounted infinite horizon Markov decision process (MDP) $\mathcal{M} = \langle \mathcal{S}, \mathcal{A}, \mathcal{P}, \rho_0, r, \gamma \rangle$, where $\mathcal{S}$ denotes the set of states, $\mathcal{A}$ the

action space, $\mathcal{P} : \mathcal{S} \times \mathcal{A} \times \mathcal{S} \to [0, 1]$ the dynamics transition function, $\rho_0 : \mathcal{S} \to [0, 1]$ the initial state distribution, and $\gamma \in [0, 1]$ the discount factor. The agent makes decisions through a policy $\pi(a|s) : \mathcal{S} \to \mathcal{A}$ and receives rewards $r : \mathcal{S} \times \mathcal{A} \to \mathbb{R}$ (or $\mathcal{S} \times \mathcal{S} \to \mathbb{R}$). For characterizing the statistical properties of a policy interacting with an MDP, the concept of occupancy measure (OM) (Syed et al., 2008; Puterman, 2014; Ho & Ermon, 2016) is proposed. Specifically, the state OM is defined as the time-discounted cumulative stationary density over the states under a given policy $\pi$: $\rho_\pi(s) = \sum_{t=0}^{\infty} \gamma^t P(s_t = s|\pi)$, similarly the state-action OM is $\rho_\pi(s, a) = \pi(a|s)\rho_\pi(s)$, and the state-transition OM is $\rho_\pi(s, s') = \int_{\mathcal{A}} \rho_\pi(s, a)\mathcal{P}(s'|s, a)\,\mathrm{d}a$. To construct a normalized density, one can take an extra normalization factor $Z = \frac{1}{1-\gamma}$ for mathematical convenience.

**Adversarial imitation learning.** Imitation Learning (IL) (Hussein et al., 2017) aims to Learn from Demonstrations (LfD), and specifically, to learn a policy from offline dataset $\mathcal{T}$ without explicitly defined rewards. Typically, the expert demonstrations consist of state-action pairs, *i.e.*, $\mathcal{T} = \{s_i, a_i\}_i$. A general IL objective minimizes the state-action OM discrepancy as $\pi^* = \arg\min_\pi \mathbb{E}_{s \sim \rho_\pi^s} \left[\ell\left(\pi_E(\cdot|s), \pi(\cdot|s)\right)\right] \Rightarrow \arg\min_\pi \ell\left(\rho_{\pi_E}(s, a), \rho_\pi(s, a)\right)$, where $\ell$ denotes some distance metric function. Adversarial imitation learning (AIL) (Ho & Ermon, 2016; Fu et al., 2017; Ghasemipour et al., 2020) is explored to optimize the objective by incorporating the advantage of Generative Adversarial Networks (GAN). In (Ho & Ermon, 2016), the policy learned by RL on the recovered reward is characterized by

$$\arg\min_\pi -H(\pi) + \psi^*(\rho_\pi - \rho_{\pi_E}),\tag{1}$$

where $\psi$ is a regularizer, and $\psi^*$ is its convex conjugate. According to Eq. (1), various settings of $\psi$ can be seen as a distance metric leading to various solutions of IL, like any $f$-divergence (Ghasemipour et al., 2020). In particular, Ho & Ermon (2016) chose a special form of $\psi$ and the second term becomes minimizing the JS divergence by training a discriminator that differentiates between the agent's state-action samples and the expert data:

$$\mathcal{L}(D) = -\mathbb{E}_{(s,a) \sim \mathcal{T}}[\log D(s, a)] - \mathbb{E}_{(s,a) \sim \pi}[\log\left(1 - D(s, a)\right)].\tag{2}$$

As the reward can be derived from the discriminator, the policy can be learned with any RL algorithm. When the action information is absent in demonstrations, *i.e.*, $\mathcal{T} = \{s_i, s_i'\}_i$, a common practice is to minimize the discrepancy of the state transition OM $\pi^* = \arg\min_\pi[\ell\left(\rho_{\pi_E}(s, s'), \rho_\pi(s, s')\right)]$ instead by constructing a discriminator for state transitions (Torabi et al., 2019):

$$\mathcal{L}(D) = -\mathbb{E}_{(s,s') \sim \mathcal{T}}[\log D(s, s')] - \mathbb{E}_{(s,s') \sim \pi}[\log\left(1 - D(s, s')\right)].\tag{3}$$

In the visual imitation learning problem we study here, visual demonstrations usually do not contain expert actions, so we shall take the state transition OM form (Eq. (3)) in the solution.

## 3 ADVERSARIAL VISUAL IMITATION WITH PATCH REWARDS

The proposed Adversarial Imitation Learning with Patch rewards (PatchAIL) method is built upon Adversarial Imitation Learning (AIL) to derive informative and meaningful patch rewards while better explaining expert demonstrations. Trained in a GAN-style, the discriminator in AIL works by differentiating the difference between agent samples and expert samples, *i.e.* through a binary-class classification problem that regards the expert data as real (label 1) and the agent data as fake (label 0). This strategy works well when learning from the low-level proprioceptive observations, where the reward is given based on the discriminator. However, this is insufficient when data is videos or image sequences. With a scalar label to classify and derive the reward simply based on the whole visual area, the images may easily be regarded as far-from-expert due to small noise or changes in local behavior. Also, there may be multiple regions of images that affect the rewarding of control tasks, while the discriminator may only focus on insufficient regions that support its overall judgment. In addition, such scalar rewards are only meaningful for the agent to learn its policy, and they lack explainability for humans to understand where the expertise comes from, such as which parts of the image contribute to higher reward or bring negative effects.

In some recent works (Cohen et al., 2021; Rafailov et al., 2021), the discriminator measures the expertise based on the latent samples encoded by the encoder trained with pixel-based RL. Albeit this

may relieve the sensitivity to local changes, it is not intuitively reasonable. When the discriminator shares the latent space with the critic, the encoder is updated only depending on the critic loss, which is the TD error that is meant to learn correct value prediction depending on the reward obtained from the discriminator. On the other hand, the discriminator is updated according to the latent space induced by the encoder. As both modules must be learned from scratch, such interdependence can cause training instability, and deteriorate the attention to key elements related to the expertise. Besides, it still lacks explainability due to the scalar label.

## 3.1 FROM SCALAR TO PATCHES

We make an intuitive and straightforward modification to the classic way of measuring expertise, *i.e.* evaluating different local parts on the given images by replacing the scalar judgment with patch ones. By evaluating local parts, on one hand, local changes only affect some of the patches rather than the whole, increasing the robustness of the overall judgment. On the other hand, the discriminator is forced to focus on more areas. Also, in this way, the classification result and reward on each patch can measure how each part of images contributes the similarity to expert visual demonstrations, which constructs a holistic view for us to understand where the expertise comes from.

**Patch discriminator.** To implement the idea of the AIL framework, we make the discriminator classify image patches and give *patches of reward* in lieu of *scalar rewards* by learning a single classification result on an image. Formally, given a visual demonstration from experts $(s, s') \sim \pi_E$, and the sample from the agent policy $\pi$, we optimize the patch discriminator $\mathbf{D}_{P \times P}$ with patch labels:

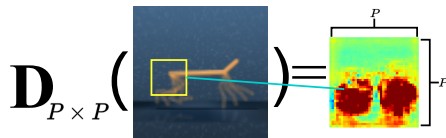

Figure 2: Illustration of the patch discriminator, where each pixel of the right-side mosaic represents the patch logit corresponding to a local region of the left-side image.

$$\mathcal{L}(\mathbf{D}) = -\mathbb{E}_{(s,s) \sim \pi_E}[\log \mathbf{D}_{P \times P}(s, s')] - \mathbb{E}_{(s,s') \sim \pi}[\log(\mathbf{1}_{P \times P} - \mathbf{D}_{P \times P}(s, s'))]. \quad (4)$$

In this formula, the discriminator outputs a matrix of probabilities $\mathbf{D}_{P \times P} : \mathcal{S} \times \mathcal{S} \to [0, 1]^{P \times P}$, where $P$ is the height/width of each image patch, as illustrated in Fig. 2. In other words, the classification loss is computed on each patch logit, rather than on the averaged logit of all patches. Furthermore, we do not have to maintain an extra encoder for the discriminator and the discriminator learns everything directly from pixels.

## 3.2 LEARNING FRAMEWORK

**Reward aggregation.** Generally, the probabilities/logits given by $D$ need to be further transformed by $R = h(D)$ to become a reward function where $h$ transforms probabilities to reward signals, *e.g.* in GAIL (Ho & Ermon, 2016) $h = \log D$ or $R = -\log(1 - D)$, and $R = \log D - \log(1 - D)$ in AIRL (Fu et al., 2017) and DAC (Kostrikov et al., 2018). In addition, after constructing patch rewards, we need an aggregation function to integrate the global information while deriving scalar signals for training RL agents. Most straightforwardly, one can choose `mean` as the aggregation operator that evaluates the overall expertise compared to expert visual demonstrations for given image sequences, whilst other kinds of operations can also be adopted. Formally, we can derive the reward function from the discriminator output as

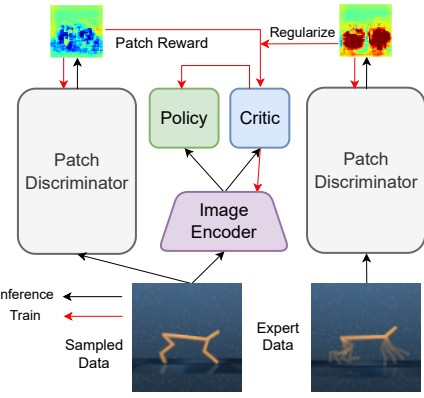

Figure 3: The framework of PatchAIL.

$$R(s, s') = Aggr(\mathbf{R}_{P \times P}(s, s')) = Aggr(h(\mathbf{D}_{P \times P}(s, s'))), \quad (5)$$

where $Aggr$ aggregates the patch rewards into scalar signals with operators `mean`, `max`, `min` and `median`. Different aggregation operators can incorporate different inductive biases into the problem and lead to various learning preferences.

**Patch regularization.** By aggregating the patch rewards into scalars, the rich information contained in the patch rewards is lost. In other words, the scalar rewards again only tell the agent about the overall judgment, and the details such as which part of the image is similar to the expert demonstration do not help the learning. To solve this issue, we further propose to regularize the *agent's* patch rewards by comparing them to the *expert's* patch rewards. Intuitively, we not only require the agent to maximize the similarity of the sampled data to the expert but also want the distribution of the agent's patch rewards to be similar to the expert's. To this end, we construct a normalized probability mass of a discrete distribution $\Delta(\mathbf{D}_{P \times P}(s, s'))$ from the output logits of the patch discriminator (for simplicity we abuse the symbol $\mathbf{D}_{P \times P}$ as the unnormalized logits by the discriminator in the following). Formally, we want to maximize the similarity $\mathrm{Sim}(s, s')$ to the demonstrations for the observation pair $(s, s')$, where the similarity $\mathrm{Sim}(s, s')$ is defined by the nearest statistical distance to the distributions induced by expert samples:

$$\max_{\pi} \ \mathbb{E}_{(s,s') \sim \pi}[\mathrm{Sim}(s, s')]$$
$$\mathrm{Sim}(s, s') = \exp\left(-\min_{(s_E, s'_E)} \ell(\Delta(\mathbf{D}_{P \times P}(s, s')), \Delta(\mathbf{D}_{P \times P}(s_E, s'_E)))\right), \quad (6)$$

where $\ell$ represents some distance metrics. However, computing Eq. (6) can be time-consuming during training. As a replacement, we choose to compute a simplified similarity $\overline{\mathrm{Sim}}(s, s')$ as the distance to the distribution induced by the averaged logits of expert samples instead:

$$\overline{\mathrm{Sim}}(s, s') = \exp\left(-\ell(\Delta(\mathbf{D}_{P \times P}(s, s')), \Delta(\mathbb{E}_{(s_E, s'_E) \sim \pi_E}[\mathbf{D}_{P \times P}(s_E, s'_E)]))\right). \quad (7)$$

In the experiment we compare these two choices and find the simplified similarity plays a good role in measuring the distance between the expert demonstrations and agent samples. Particularly, if we instantiate $\ell$ as KL divergence $D_{\mathrm{KL}}$, the similarity is then computed as

$$\overline{\mathrm{Sim}}(s, s') = \exp\left(-\sum_{i=1}^{P} \sum_{j=1}^{P} \Delta_{i,j}(\mathbf{D}_{P \times P}(s, s')) \log \frac{\Delta_{i,j}(\mathbf{D}_{P \times P}(s, s'))}{\mathbb{E}_{(s_E, s'_E) \sim \pi_E}[\Delta_{i,j}(\mathbf{D}_{P \times P}(s_E, s'_E))]}\right), \quad (8)$$

where $\Delta_{i,j}$ denotes the probability value of the $(i, j)$-th entry of $\Delta(\mathbf{D}_{P \times P})$. The similarity can serve as the regularizer to weigh the aggregated reward signals. Considering ways of using the similarity regularizer, we have two variants of the PatchAIL algorithm. The first version is named *PatchAIL-W(eight)*, where we take the regularizer as the weighting multiplier on the reward:

$$R(s, s') = \lambda \cdot \overline{\mathrm{Sim}}(s, s') \cdot Aggr(h(\mathbf{D}_{P \times P}(s, s'))) ; \quad (9)$$

the other variant is named *PatchAIL-B(onus)*, where the regularizer serves as a bonus term:

$$R(s, s') = \lambda \cdot \overline{\mathrm{Sim}}(s, s') + Aggr(h(\mathbf{D}_{P \times P}(s, s'))) . \quad (10)$$

$\lambda$ is the hyperparameter for balancing the regularization rate. In our experiments, we find that such a regularization helps stabilize the training procedure on some tasks, as shown in Fig. 4.

**Implementation of the patch discriminator.** In principle, the architecture of the patch discriminator is an open choice to any good designs to construct patch rewards, such as vision transformer (Isola et al., 2017) (explicitly break input images as patches) or fully-convolution network (FCN) (Long et al., 2015) (output matrix implicitly mapped back to patches on input images). In this paper, we choose to implement the patch discriminator as a 4-layer FCN, referring to the one in (Zhu et al., 2017). In detail, the discriminator has four convolution layers without MLP connections and keeps the last layer as Sigmoid to give the probabilities of the classification for each patch, where each patch may share some perceptive field with each other. The discriminator classifies if each patch is real (*i.e.*, expert) or fake (*i.e.*, non-expert). As a result, we can obtain multi-dimensional rewards by evaluating the expertise of every local region of images.

**Overall algorithm.** In a nutshell, PatchAIL obtains patch rewards from the patch discriminator, and training agents with reward aggregation and regularization. The same as other AIL methods, PatchAIL works with any off-the-shelf pixel-based RL algorithm. In this paper, we choose DrQ-v2 (Ye et al., 2020) as the underlying component for learning the policy and integrating gradient penalty (Gulrajani et al., 2017) for training the discriminator. The algorithm is trained under an off-policy style inspired by Kostrikov et al. (2018), therefore the agent samples are from the replay buffer $\mathcal{B}$ and the expert samples are from the demonstration dataset $\mathcal{T}$. An overview of the framework is illustrated in Fig. 3, and the step-by-step algorithm can be found in Appendix A.

## 4 RELATED WORK

**Pixel-based reinforcement learning.** The proposed solution for Visual Imitation Learning (VIL) is built on the recent success of pixel-based reinforcement learning. When learning from raw pixels, a good state representation of raw pixels encoded by image encoders is crucial to the learning performance, and it is widely acknowledged that the policy shares the latent representation with the critic, and only the critic loss is used for updating the encoder (Yarats et al., 2019; Laskin et al., 2020a;b; Yarats et al., 2020). Besides, many recent works integrate diverse representation learning techniques for pixel-based RL. For instance, Yarats et al. (2019) trained a regularized autoencoder jointly with the RL agent; Lee et al. (2020) learned a latent world model on top of VAE features and built a latent value function; Laskin et al. (2020a) optimized contrastive learning objectives to regularize the latent representations; Laskin et al. (2020b); Yarats et al. (2020; 2021) utilized data augmentation for efficient representation learning. For learning such image encoders, all these methods require a well-defined reward function, which is inaccessible for VIL.

**Visual imitation learning.** In the imitation learning community, most previous solutions are designed for state-based settings, where the states/observations are low-level perceptive vector inputs, such as (Ho & Ermon, 2016; Kostrikov et al., 2018; 2019; Liu et al., 2021; Reddy et al., 2019; Brantley et al., 2019; Garg et al., 2021; Liu et al., 2022). Among them, some can be directly used for visual imitation. For example, Reddy et al. (2019) allocated a positive reward (+1) to expert data and a negative reward (-1) to experience data sampled by the agent; Brantley et al. (2019) used the disagreement among a set of supervised learning policies to construct the reward function. Instead of learning the reward function, Garg et al. (2021) chose to learn a Q function from expert data directly, but only tested on discrete Atari games. Besides, Pomerleau (1991); Pathak et al. (2018) designed supervised learning techniques for resolving real-world imitation learning tasks. Recently, Rafailov et al. (2021); Cohen et al. (2021) extended previous adversarial imitation learning solution based on latent samples with existing pixel-based RL algorithms was intuitively unreasonable as discussed in Section 3. Based on Cohen et al. (2021), Haldar et al. (2022) further augmented a filtered BC objective into the imitation learning procedure, showing a considerable improvement in efficiency and final performance. However, their work requires access to experts' action information, which is usually absent in image-based demonstrations, where we only have visual observations. In Appendix B.3 we also report experiments of *PatchAIL* augmented with BC objective assuming the accessibility to the action information, reaching much better efficiency and more stable learning performance. Besides the learnability, none of these works can recover explainable reward signals that interpret the expertise of the demonstration and help people to understand the nature of tasks.

**Image patches in literature.** Breaking up images into patches is not a novel idea in the deep learning community, not even the proposal of the patch discriminator, as it has already been used in Dosovitskiy et al. (2020); Zhu et al. (2017); Isola et al. (2017). On one hand Li & Wand (2016); Isola et al. (2017); Zhu et al. (2017) argued that using a patch discriminator can better help the generator learn details, where Li & Wand (2016) used it to capture local style statistics for texture synthesis, and Isola et al. (2017); Zhu et al. (2017) used it to model high-frequencies; on the other hand, Dosovitskiy et al. (2020) proposed to utilize the ability of transformer (Vaswani et al., 2017) by splitting the image into patches, mainly for capturing the semantic information in images. Different from them, our usage of image patches along with the patch discriminator is distinct and novel in the context of RL and VIL, as we are seeking a fine-grained expertise measurement of the images given a set of visual demonstrations, like where the agent should be rewarded and where not.

## 5 EXPERIMENTS

In our experiments, we thoroughly examine our proposed Adversarial Imitation Learning with Patch rewards (PatchAIL) algorithm. We first conduct a fair comparison against baseline methods on a wide range of pixel-based benchmarking tasks on DeepMind Control Suit (Section 5.1), showing the effectiveness of PatchAIL; next, we compare the spatial attention maps given by PatchAIL with baselines to show the reasonability of the reward given by PatchAIL; finally, we conduct a set of ablation studies to investigate the effect of different choices in PatchAIL. Due to the page limit, we leave more results on Atari tasks, additional ablation experiments and more implementation details in Appendix B.

Table 1: Averaged returns on 7 DMC tasks at 500K and 1M environment steps over 5 random seeds using 10 expert trajectories. gray columns represent our methods, and *Shared-Enc. AIL* is a solution proposed in (Cohen et al., 2021). PatchAIL and its variants outperform other approaches in data-efficient (500K) and asymptotic performance (1M). Since BC does not need online training, we report their asymptotic performance.

| *500K step scores* | PatchAIL w.o. Reg. | PatchAIL-W | PatchAIL-B | Shared-Enc. AIL | Independent-Enc. AIL | BC | Expert |
|---|---|---|---|---|---|---|---|
| Cartpole Swingup | **838±14** | 784±15 | 811±23 | 729±40 | 693±54 | 521±120 | 859±0 |
| Cheetah Run | 413±16 | 409±29 | 415±30 | **533±34** | 335±23 | 185±49 | 890±19 |
| Finger Spin | 850±32 | **871±25** | 839±23 | 777±48 | 547±51 | 284±120 | 976±9 |
| Hopper Stand | 235±128 | 430±99 | 207±107 | 206±89 | **729±38** | 386±72 | 939±9 |
| Walker Stand | 896±38 | **902±34** | 900±39 | 881±15 | 865±20 | 496±70 | 970±20 |
| Walker Walk | 521±103 | 513±117 | **542±60** | 200±5 | 455±92 | 566±110 | 961±18 |
| Quadruped Run | 230±38 | 239±43 | **273±17** | 220±33 | 265±34 | 277±58 | 547±136 |
| *1M step scores* | | | | | | | |
| Cartpole Swingup | **854±2** | 844±9 | 833±25 | 844±3 | 367±97 | 521±120 | 859±0 |
| Cheetah Run | 654±24 | 681±21 | 691±15 | 646±19 | **791±13** | 185±49 | 890±19 |
| Finger Spin | 875±38 | 855±33 | **943±7** | 826±71 | 110±54 | 284±120 | 976±9 |
| Hopper Stand | 781±37 | 693±47 | **803±19** | 676±130 | 762±20 | 386±72 | 939±9 |
| Walker Stand | 916±41 | **973±2** | 972±2 | 886±27 | 824±25 | 496±70 | 970±20 |
| Walker Walk | 956±3 | 951±3 | **960±2** | 952±2 | 945±5 | 566±110 | 961±18 |
| Quadruped Run | 406±14 | **423±9** | 416±8 | 331±33 | 354±34 | 277±58 | 547±136 |

**Baselines.** Our experiments cover three kinds of baseline methods.

- Shared-Encoder AIL: the discriminator learns from latent samples as discussed in Section 3, which is also the proposal in Cohen et al. (2021);

- Independent-Encoder AIL: the discriminators learn from images directly with a scalar label.

- Behavior cloning (BC): supervised learning for the policy function, which is included only for reference as it gets access to the expert's exact actions.

For a fair comparison, all baselines are implemented under the same code base with the same architecture of the policy and Q networks. *AIL* methods take the same choice of the reward transformation function $R = \log D - \log(1-D)$ as in Fu et al. (2017); Kostrikov et al. (2018). We emphasize that all agents *except* BC can only require image sequences and cannot obtain accurate action information.

**Experimental setup.** Our experiments are conducted on 7 tasks in the DeepMind Control (DMC) Suite (Tassa et al., 2018). For expert demonstrations, we use 10 trajectories collected by a DrQ-v2 (Yarats et al., 2021) agent trained using the ground-truth reward and collect. We follow Yarats et al. (2021) to use `Random-Shift` to augment the input samples for training the policy, critic and discriminator for every method tested. The default choice of aggregation function for PatchAIL is `mean`; and the default architecture of the patch discriminator is a 4-layer FCN with kernels [4×4, 32, 2, 1], [4×4, 64, 1, 1], [4×4, 128, 1, 1] and [4×4, 1, 1, 1] (represented as [size, channel, stride, padding]), whose output has $39 \times 39$ patches in total and each has a receptive filed of 22. We put results trained by 10 trajectories in the main text and leave the rest in Appendix B. Since the expert can be seen as the optimal policy under the ground-truth reward function, *i.e.*, its trajectory distribution regarded as proportional to the exponential return (Ho & Ermon, 2016; Liu et al., 2021), we take the ground-truth return as the metric for measuring the agents. Throughout experiments, these algorithms are tested using *fixed* hyperparameters on *all* tasks and the full set of hyperparameters is in Appendix B. All methods use $2,000$ exploration steps which are included in the overall step count. We run 5 seeds in all experiments under each configuration and report the mean and standard deviation during the training procedure.

## 5.1 IMITATION EVALUATIONS

We compare our proposed PatchAIL and its variants to all baseline methods on 7 DeepMind Control Suit tasks. In Tab. 1, we present the average returns on each task at both 500K and 1M training steps, and in Fig. 4, we show the learning curve of each method. Statistics from Tab. 1 indicate that on almost all testbeds, PatchAIL methods own better imitation efficiency and efficacy. At 500K frames, PatchAIL obtains an averaged performance gain of 44% over Shared-Encoder AIL and 13% over Independent-Encoder AIL; at 1M steps, the averaged performance gain over Shared-Encoder AIL becomes 11% and over Independent-Encoder AIL is 131%. From Fig. 4, we find that the encoder-based AIL baselines can work well on various tasks. Nevertheless, by learning its representation, independent-encoder AIL is rather unstable in almost all tasks during the learning procedure. Specifically, when it reaches a near-expert performance, it starts to fall and becomes worse (such as

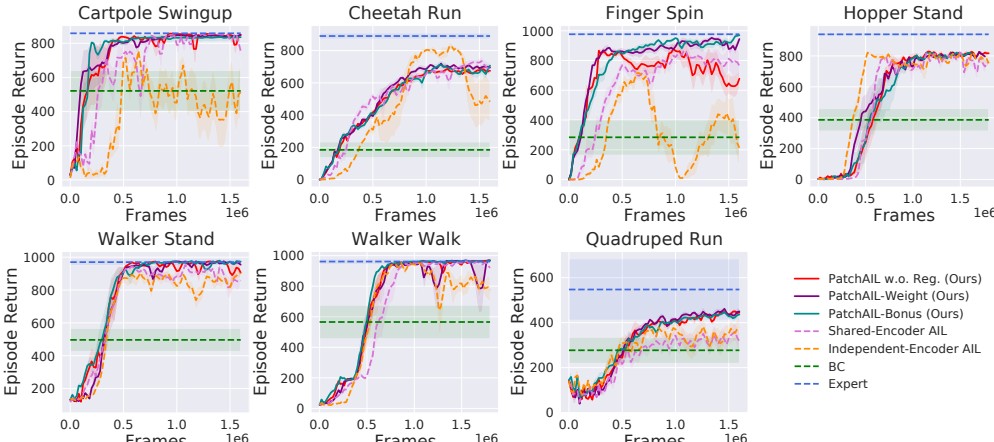

Figure 4: Learning curves on 7 DMC tasks over 5 random seeds using 10 expert trajectories. The curve and shaded zone represent the mean and the standard deviation, respectively.

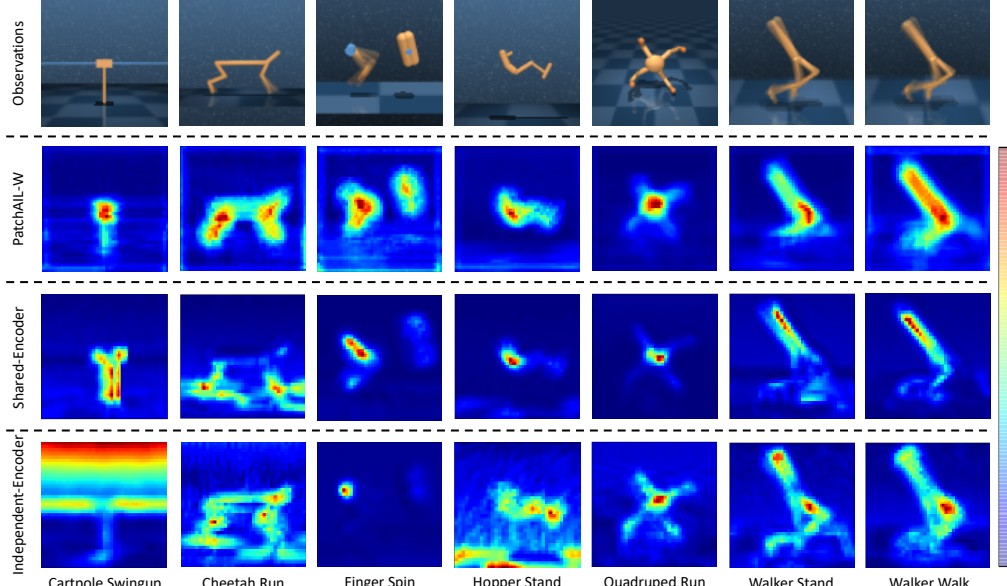

Figure 5: Selected spatial attention maps showing where discriminators focus on to make a decision, given by discriminators of PatchAIL-W, shared-encoder AIL and independent-encoder AIL on 7 Deepmind Control Suite tasks using the final model trained around 200M frames. Red represents high value while blue is low.

Cartpole-Swingup, Cheetah-Run, Finger-Spin, Walker-Stand, and Walker-Walk). On the contrary, shared-encoder AIL performs better and tends to become steady at the late stage of training, where the state representation is almost unchanged, showing it can be effective in many cases. However, in some tasks, it has less learning efficiency in the early learning stage due to the interdependent update of the discriminator and the critic (such as Cartpole-Swingup, Finger-Spin and Walker-Walk). In comparison, PatchAIL and its variants are more stable and much more efficient. This is obvious on Cartpole-Swingup and Finger-Spin. Notably, they have a noticeable performance advantage in the Quadruped-Run domain. Moreover, with the proposed patch regulation, PatchAIL becomes more stable, which is quite evident in the Finger-Spin domain, yet the main performance gain comes from the patch reward structure given the overall results.

## 5.2 VISUAL EXPLANATIONS

In this part, we focus on the visual explanations, which we deem as a superior advantage of PatchAIL compared with baselines. To understand where the discriminators focus, we visualize the spatial feature maps of different methods in Fig. 5. It is clear that PatchAIL provides complete and holistic concentration on the key elements of images. Particularly, we notice that the two encoder-based

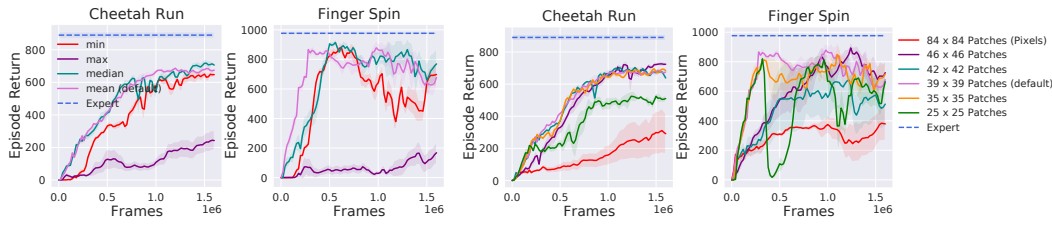

(a) Ablation study for aggregation functions.      (b) Ablation study for the number of patches.

Figure 6: Ablation studies, only tested with PatchAIL w.o. Reg.

baselines both miss key areas on some tasks, such as on Finger-Spin, where both methods seem mainly to make judgments on the left finger. Furthermore, shared-encoder AIL also overlooks the leg on Hopper-Stand and Quadruped-Run; and independent AIL pays more attention to the background on Cartpole-Swingup, Cheetah-Run and Quadruped-Run. These observations meet our assumptions in Section 3. To better comprehend and explain which part of images contribute to the overall expertise, we mapped each patch reward back to its corresponding pixels, weighted by the attention map, on both expert data and random data. The results have been illustrated in Section 1, which indicates where the high rewards or low ones come from.

## 5.3 ABLATION STUDIES

**Ablation on aggregation function.** In the proposed PatchAIL framework, we learn patch rewards for image data and then aggregate them to obtain a scalar reward for learning the policy by RL algorithms. In this paper, the default choice of the aggregation function is `mean`, but there leaves a wide range of choices. Therefore we conduct ablation experiments on various choices of aggregation functions, like `min`, `max` and `median`. The results on the Finger-Spin and Cheetah-Run domain are shown in Fig. 6a. It can be seen that `mean` is one-of-the-best choice among these aggregation functions, while `median` and `min` also preserve well learnability. On the other hand, the `max` operator tends to fail in both domains, which may be attributed to its over-optimism of bad data.

**Ablation on size of patches.** We are also interested in the impact of the size of patches along with the receptive field of each patch on the performance of PatchAIL. As mentioned above, the default implementation of PatchAIL utilizes a 4-layer FCN, and each layer uses a convolution kernel with a size of 4×4, resulting in 39×39 patches in the last layer. Each of them has a 22×22 reception field. In ablation, we only modify the kernel size as $[2, 3, 5, 8]$ with $[46 \times 46, 42 \times 42, 35 \times 35, 25 \times 25]$ patches and reception field sizes of $[8 \times 8, 15 \times 15, 29 \times 29, 50 \times 50]$, keeping the stride and padding size the same as the default setting; besides, we also test a special case where each layer of the FCN is a $1 \times 1$ convolution layer, resulting in a pixel level patch reward (84×84 patches and each has a $1 \times 1$ reception field). The learning curves on the Finger-Spin and Cheetah-Run domain are illustrated in Fig. 6b, where we observe three apparent phenomena: 1) PatchAIL with bigger reception fields behaves with high training efficiency in the early stage; 2) with the biggest reception fields (*i.e.*, $25 \times 25$ patches with a kernel size of $8 \times 8$) PatchAIL can be extremely unstable; 3) learning rewards for each pixel fails to obtain a well-performed policy; 4) some tasks like Cheetah-Run are less sensitive to the change of patch size than Finger-Spin, which maybe can explain why PatchAIL almost shows no advantage in Fig. 4 compared to baselines.

## 6 CONCLUSION

We proposed PatchAIL, an intuitive and principled learning framework for efficient visual imitation learning (VIL). PatchAIL offers explainable patch-level rewards, which measure the expertise of various parts of given images and use this information to regularize the aggregated reward and stabilize the training, in place of less informative scalar rewards. We evaluate our method on the widely-used pixel-based benchmark DeepMind Control Suite. The results indicate that PatchAIL supports efficient training that outperforms baseline methods and provides valuable interpretations for VIL. We think that the proposed patch reward is a crucial method for utilizing the wealth of information contained in image observations, and that there is still a great potential for leveraging the patch-level information to enhance visual reinforcement learning and visual imitation learning.

## 7 ETHICS STATEMENT

This submission adheres to and acknowledges the ICLR Code of Ethics and we make sure that we do not violate any ethics concerns.

## 8 REPRODUCIBILITY STATEMENT

All experiments included in this paper are conducted over 5 random seeds for stability and reliability. The algorithm outline is included in Appendix A, and the hyperparameters and implementation details are included in Appendix C. Our code has been released to the public.

## ACKNOWLEDGEMENT

The SJTU Team is supported by Shanghai Municipal Science and Technology Major Project (2021SHZDZX0102) and National Natural Science Foundation of China (62076161). Minghuan Liu is also supported by Wu Wen Jun Honorary Doctoral Scholarship, AI Institute, SJTU. We sincerely thank Quanhong Fu very much for her careful revising of our first manuscripts, and we also thank Qingfeng Lan, Zhijie Lin, Xiao Ma for helpful discussions.

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

# Appendices

## A  ALGORITHM OUTLINE

---

**Algorithm 1** Adversarial Imitation Learning with Patch rewards (PatchAIL)

---

1: **Input:** Empty replay buffer $\mathcal{B}$, patch discriminator $\mathbf{D}_{P \times P}$, policy $\pi$, expert demonstrations $\mathcal{T}$.
2: **for** $k = 0, 1, 2, \cdots$ **do**
3:     Collect trajectories $\{(s, a, s', \text{done})\}$ using current policy $\pi$
4:     Store $\{(s, a, s', \text{done})\}$ in $\mathcal{B}$
5:     Sample $(s, a, s') \sim \mathcal{B}, (s_E, s'_E) \sim \mathcal{D}$
6:     Update the discriminator $\mathbf{D}_{P \times P}$ followed by Eq. (4)
7:     Compute the reward $r$ for $(s, a, s')$ with the patch discriminator $\mathbf{D}_{P \times P}$ by Eq. (5) or Eq. (9) or Eq. (10)
8:     Update the policy $\pi_\theta$ by any off-the-shelf RL algorithm
9: **end for**

---

## B  ADDITIONAL EXPERIMENT RESULTS

### B.1  THE CHOICE OF OBSERVATION PAIRS

In this section, we test the choice of using an observation $s$ or observation pairs $(s, s')$ for the discriminator in our experiments. The learning curve of the shared-encoder AIL and PatchAIL is shown in Fig. 7. It seems that this choice makes nearly no difference for shared-encoder AIL, as it learns from latent samples. On the contrary, utilizing observation pairs $(s, s')$ shows better performance for PatchAIL. In experiments, since the visual observation is always composed of a stack of images (e.g., a stack of 3 frames in DeepMind Control Suite used in this paper), using an observation pair is just stacking two stacks of images into one bigger stack.

### B.2  IMITATION USING FEWER DEMONSTRATIONS

We also evaluate our algorithm against baselines using fewer demonstrations. As shown in Fig. 8, we can draw consistent conclusions with the results using 10 trajectories on most of tasks. Nevertheless, with less demonstrations, PatchAIL and its variants appear to have no advantage on Cartpole-Swingup with fewer demonstrations, they emerge as winners on Hopper-Stand.

### B.3  AUGMENTED WITH BC WITH ONLY 1 DEMONSTRATION

When we can get access to the action information, we are able to further improve the imitation performance by mimicking the experts. Motivated by Haldar et al. (2022) and Jena et al. (2021), we test the performance by augmenting PatchAIL with BC, where the weight of BC loss is annealing following Jena et al. (2021). We compare with the recent state-of-the-art work ROT (Haldar et al., 2022), and the two encoder-based baselines in Section 5, and show the results in Fig. 9. We find that

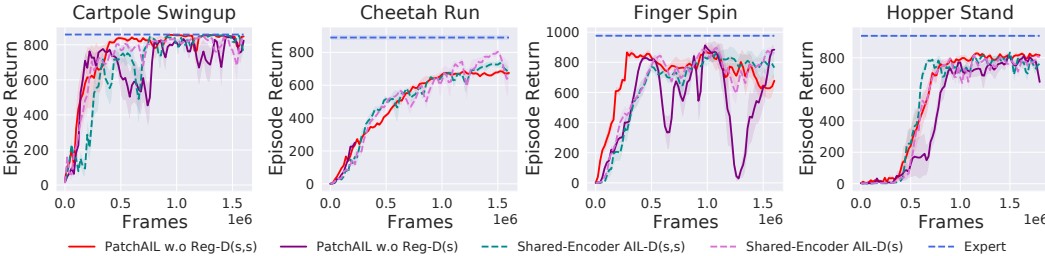

Figure 7: Ablation for the choice of observation pairs over 5 random seeds.

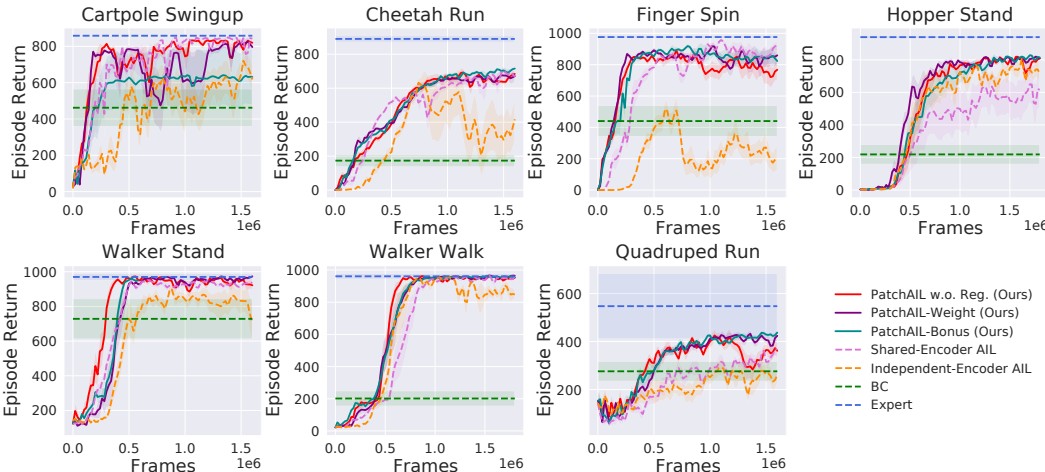

Figure 8: Learning curves on 7 DMC tasks over 5 random seeds using 5 expert trajectories. The observations are almost consistent with the results using 10 trajectories except on Cartpole-Swingup and Hopper-Stand.

PatchAIL and its variants show faster convergence speed but ROT seems to have a slightly better final performance, while PatchAIL-W and PatchAIL-B are able to alleviate the unstable problem of PatchAIL. In comparison, encoder-based baselines does not show any advantage when augmented with BC.

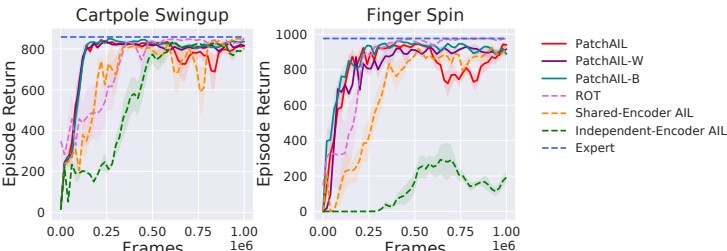

Figure 9: Learning curves by augmenting PatchAIL with annealing BC using 1 expert trajectories over 5 random seeds. PatchAIL, PatchAIL-W and PatchAIL-B all show faster convergence speed than all baselines but ROT seems to have a slightly better final performance.

## B.4 COMPARING THE SIMILARITY FORMULATION

In Section 3.2, we propose a similarity $\texttt{Sim}(s, s')$ defined by the nearest statistical distance to the distributions induced by expert samples (will denote as Eq. (6)), and a simplified version $\overline{\texttt{Sim}}(s, s')$ as the distance to the distribution induced by the averaged logits of expert samples for computing efficiency (will denote as Eq. (7)). In this subsection, we compare the effect of these two different similarity version on two representative tasks, Cartpole-Swingup and Finger-Spin using 10 and 5 expert trajectories. The learning results and corresponding averaged similarities during training are shown in Fig. 10. In our experiment, we let all similarity begins from the same initial value (shown in Tab. 2) to eliminate the effect of different value scale. From Fig. 10, we observe two facts: 1) Eq (7) and Eq (6) achieve similar similarity trends and Eq (6) obtains larger similarity values in average 2) Eq (6) makes learning less aggressive, leading to slower initial learning (as on Finger-Spin), but can be more stable and may converge to better ultimate results, especially when there are fewer expert data.

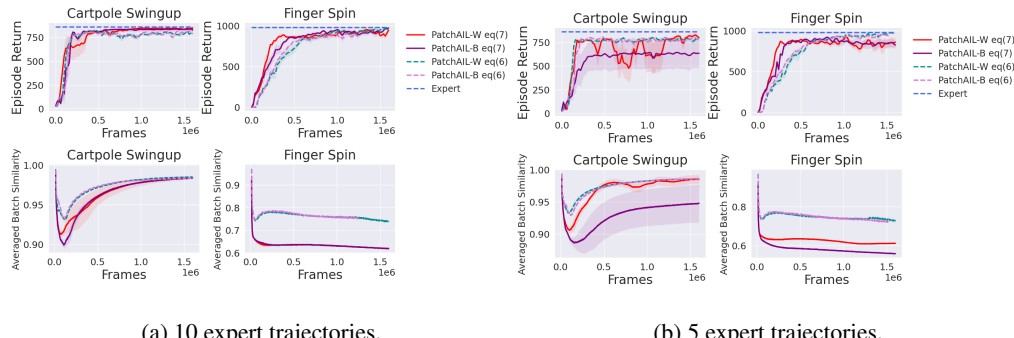

(a) 10 expert trajectories.        (b) 5 expert trajectories.

Figure 10: Comparing different similarity formulation Eq (7) and Eq (6).

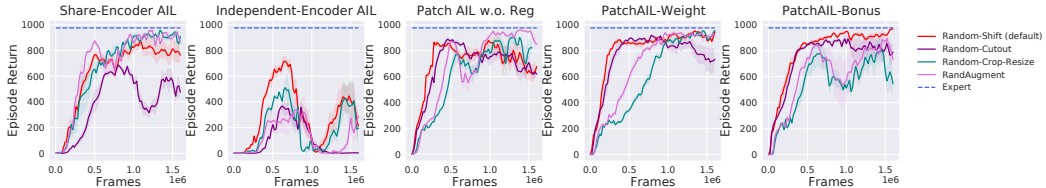

Figure 11: Different data augmentation strategies for discriminators on Finger-Spin with 10 expert trajectories over 5 random seeds.

### B.5 DIFFERENT DATA AUGMENTATION STRATEGIES FOR DISCRIMINATORS

In our claim, we argue that the discriminator of the encoder-based solutions can be sensitive to changes in local behavior, and may only focus on insufficient regions that support its overall judgment. In our experiment, we do observe that both shared-encoder AIL and independent-encoder AIL lost come regions of given image samples. However, in the deep learning and computer vision community, researchers have developed various kinds of data augmentation methods to alleviate such a problem by improving the robustness of the discriminator. Although in our experiments, we have included a `Random-Shift` augmentation for the training of the discriminator, which has been shown to be the most useful data augmentation technique in pixel-based reinforcement learning problems (Kostrikov et al., 2019; Laskin et al., 2020b). We still wonder if different data augmentation strategies can improve VIL performance. To verify this, we try `Random-Cut-Out`, `Random-Crop-Resize`, and `RandAugment` (Cubuk et al., 2020) (without color operations since we are augmenting stack of images) strategy using the two encoder-based baselines on the Finger-Spin domain, which we think may be the most useful augmentation strategy for the discriminator to differentiate samples on the DeepMind Control Suite testbed. The results are shown in Fig. 11. Compared to observations in Fig. 4, it is easy to conclude that augmentation strategies are not the key to making independent-encoder AIL successful, but patch rewards can make discriminator works directly on pixels; in comparison, `Random-Crop-Resize` and `Random-Shifts` lead share-encoder AIL into similar training efficiency, while `RandAugment` may promote the efficiency and final performance, causing more instability. And it turns out the best augmentation strategy for PatchAIL and its variants is the default `Random-Shift`, leading to stable, efficient, and better performance. Nevertheless, we can still conclude the effectiveness of learning with patch rewards. But this also motivates us that improving the data-augmentation strategy with shared-encoder AIL may be worth further investigation.

### B.6 VISUAL ATTENTION WITH GRADCAM

We also try interpreting the rewards given by discriminators using GradCAM Selvaraju et al. (2017), a popular visual explanation tool. Specifically, we use GradCAM to generate the response score regarding the expert label and the non-expert label for each method. The comparisons are in Fig. 12. We find the results are mainly hard to interpret, indicating that such a tool may not be appropriate to explain the reward provided by AIL methods. But the proposed patch rewards can naturally provide

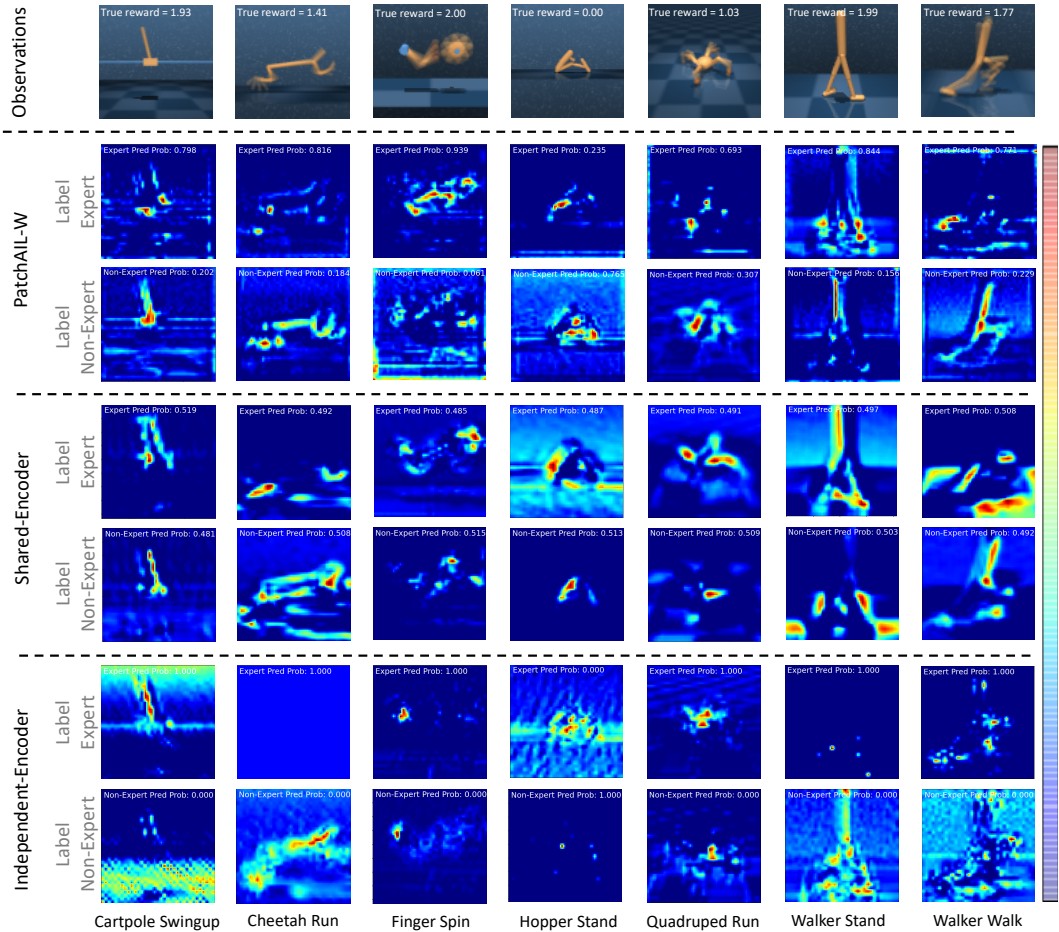

Figure 12: Visual interpretation by GradCAM for each method. PatchAIL and shared-encoder AIL show an advantage over independent-encoder in terms of focusing more on the robot and focusing less on the background. This is especially obvious in five out of seven environments (Cartpole-Swingup, Cheetah-Run, Quadruped-RUn, Walker-Stand and Walker-Walk).

interpretability. We can only observe from Fig. 12 that PatchAIL and shared-encoder AIL show advantages over independent-encoder in terms of focusing more on the robot and focusing less on the background. This is especially obvious in five out of seven environments (Cartpole-Swingup, Cheetah-Run, Quadruped-Run, Walker-Stand and Walker-Walk).

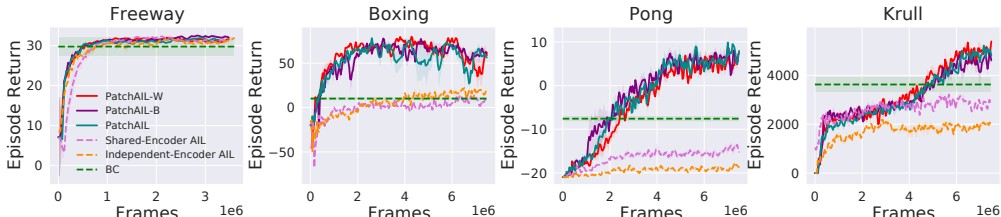

Figure 13: Learning curves on 4 Atari tasks over 5 random seeds using 20 expert trajectories. The curve and shaded zone represent the mean and the standard deviation, respectively.

### B.7 ATARI EXPERIMENTS

Except for DMC tasks, we test our methods on another commonly used pixel-based reinforcement learning benchmark, Atari. We use the data collected by RLUnplugged Gulcehre et al. (2020) as expert demonstrations. In particular, we choose 20 episodes sampled by the last 3 checkpoints for imitation without using any reward and action information (except BC). We utilize Dueling DQN Wang et al. (2016) as our base learning algorithm, whose rewards are provided by discriminators. We test all mentioned PatchAIL methods against baseline methods on four tasks, including Freeway, Boxing, Pong and Krull, and show the learning curves in Fig. 13. We train each method in less than 10M frames if it is converged. The conclusion remains consistent compared with those on DMC tasks. Concretely, PatchAIL methods are obviously observed to have better learning efficiency on all tasks, and notably have strong improvement on harder tasks.

## C IMPLEMENTATION DETAILS

### C.1 BASELINE ARCHITECTURES

The architecture of two encoder-based Adversarial Imitation Learning (AIL) baselines and the proposed PatchAIL solution are illustrated in Fig. 14. As for the detailed encoder architecture, all methods use the default structure for the policy/critic that works in Haldar et al. (2022): a 4-layer CNN with kernels [3×3, 32, 2, 0], [3×3, 32, 1, 0], [3×3, 32, 1, 0], [3×3, 32, 1, 0] (represented as [size, channel, stride, padding]), resulting in a $35 \times 35$ feature map and each entry has a receptive field of 15. Note that the encoder-based baselines also utilize such an encoder for the discriminator.

After encoding, the feature maps are flattened as a 39200-length vector, which is then put into a 4-layer MLP actor as and a 4-layer critic. For encoder-based baselines, the discriminator also follows the structure in Haldar et al. (2022) as a 3-layer MLP which outputs a scalar result. Regarding PatchAIL methods, the discriminator used for DMC domain is a 4-layer FCN with default kernels [4×4, 32, 2, 1], [4×4, 64, 1, 1], [4×4, 128, 1, 1] and [4×4, 1, 1, 1] (represented as [size, channel, stride, padding]), resulting in $39 \times 39$ patches and each has a receptive field of 22. As for Atari domain, the discriminator is a 4-layer FCN with default kernels [4×4, 32, 2, 1], [4×4, 64, 2, 1], [4×4, 128, 1, 1] and [4×4, 1, 1, 1] (represented as [size, channel, stride, padding]), resulting in $19 \times 19$ patches and each has a receptive field of 34. In our experience, the discriminators should have a reasonable structure (i.e., the resulting patch number and the receptive field size of each patch relative to the task) to achieve good results.

### C.2 IMPORTANT HYPERPARAMETERS

We list the key hyperparameters of AIL methods used in our experiment in Tab. 2. The hyperparameters are the same for all domains and tasks evaluated in this paper.

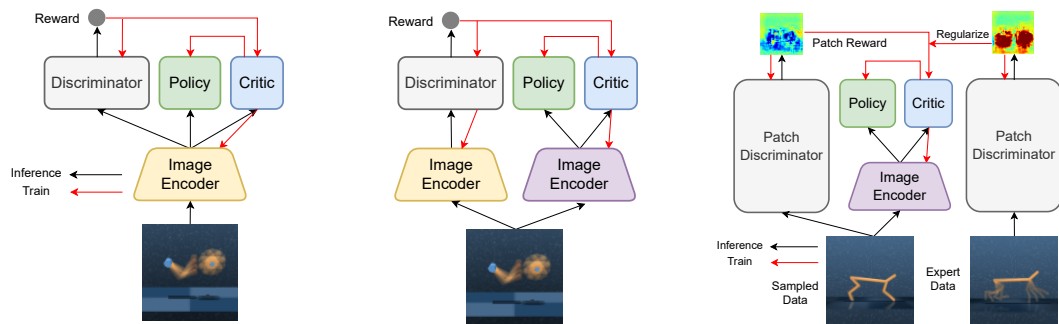

(a) Shared-encoder AIL structure.  (b) Independent-encoder AIL structure.  (c) PatchAIL structure.

Figure 14: Architectures for the two visual imitation learning baselines (*not our proposal*) and the proposed method. (a) The discriminator shares the encoder with the policy and the critic. The encoder of the policy/critic is only updated by the critic loss. (b) The encoder of the discriminator is independent of the policy/critic. (c) PatchAIL, where the discriminator is also independent to the critic's encoder.

| Method | Parameter | Value |
|---|---|---|
| Common | Default replay buffer size | 150000 |
|  | Learning rate | $1e^{-4}$ |
|  | Discount $\gamma$ | 0.99 |
|  | Frame stack / $n$-step returns | 3 |
|  | Action repeat | 2 |
|  | Mini-batch size | 256 |
|  | Agent update frequency | 2 |
|  | Critic soft-update rate | 0.01 |
|  | Feature dim | 50 |
|  | Hidden dim | 1024 |
|  | Optimizer | Adam |
|  | Exploration steps | 2000 |
|  | DDPG exploration schedule | linear(1,0.1,500000) |
|  | Gradient penalty coefficient | 10 |
|  | Target feature processor update frequency(steps) | 20000 |
|  | Default Reward scale | 1.0 |
|  | Default data augmentation strategy | random shift |
| PatchAIL-W | $\lambda$ initial value | 1.3 |
|  | Replay buffer size | 1000000 |
| PatchAIL-B | $\lambda$ initial value | 0.5 |
|  | Replay buffer size | 1000000 |
|  | Reward scale | 0.5 |

Table 2: List of hyperparameters for all domains.

