# OpenReview forum: "Visual Imitation Learning with Patch Rewards"
_ICLR.cc/2023/Conference — ICLR 2023 poster_

### Official Review · Reviewer_AJEj · 2022-10-25

**Confidence:** 5
**Correctness:** 3
**Technical Novelty And Significance:** 2
**Empirical Novelty And Significance:** 3
**Recommendation:** 5

**Clarity, Quality, Novelty And Reproducibility:**

The paper is clearly written and easy to follow. However, the proposed method is highly similar to previous AIL methods, with the main difference being that images are replaced with patches.

**Details Of Ethics Concerns:**

N/A.

**Strength And Weaknesses:**

Advantages
* The method is straightforward and intuitively corresponds to the observation that comparing patches in image observations with the expert demonstration can provide useful learning signals.
* The method is evaluated on standard pixel-based benchmarks and achieves comparable performance compared to its baselines.
* The paper offers a discussion on the visual interpretation of the proposed reward function.

Weaknesses
* The proposed method is highly similar to previous AIL methods. This itself is not a problem if it provides large performance gain. However, gain is small compared to Shared-Encoder AIL according to Figure 4, while introducing several extra parameters such as aggregation function class and patch size.
* The other claimed advantage of the method is visual interpretability. However, according to Figure 5, both Shared-Encoder as will as the proposed method have their attention maps focusing on the agent.

Others
* There are two typos in equation (4). It seems to be (s, s') ~ \pi and (s, s') ~ \pi_E.


**Summary Of The Paper:**

The paper proposes a general class of reward function defined by image patches in observations, which enables effective learning guidance when training an VIL agent, visual interpretability, and better training stability. The method is evaluated on a range of DeepMind Control tasks and achieves favorable performance compared baseline methods.

**Summary Of The Review:**

The method offers a simple AIL framework for training pixel-based RL agents. The main concern is that the empirical benefits, both in terms of asymptotic performance and sample efficiency, is not high compared to existing works. The paper claims that the method offers better visual explanations, but this particular point is not well-supported by Figure 5 where the baseline method already provides a similar level of interpretability.

--- Post Rebuttal ---
After the rebuttal responses and reading other reviewers' comments, I increased my score from 3 to 5 with more clarifications from the authors. My main concern is that the asymptotic performance gain over the baselines is specific to particular environments, and the visualization analysis is spuriously correlated with motion and does not distinguish positive and negative contributions to the reward.

---

> ### Author Response · Authors · 2022-11-13
> **Author Reply to Reviewer AJEj**
>
> Thanks for your helpful feedback to improve our manuscript! We have revised our typos in the revision and tried to address your concerns as below.
> ***
> **Q1**: The proposed method is highly similar to previous AIL methods. This itself is not a problem if it provides a large performance gain. However, the gain is small compared to Shared-Encoder AIL according to Figure 4, while introducing several extra parameters such as aggregation function class and patch size.
>
> **A1**: We sincerely argue that the performance gain is not tiny but significant. To be specific, at 500K frames, in the Walker-Walk task, the performance gain of PatchAIL over Shared-Encoder AIL is **171%** and the gain over Independent-Encoder AIL is **19%**. At 1M steps, in the Finger-Spin task, the performance gain of PatchAIL over Shared-Encoder AIL is **14%** and the gain over Independent-Encoder AIL is **757%**.
>
> As for the averaged performance gain across 7 environments, at 500K frames, PatchAIL obtains an average performance gain of **44%** over Shared-Encoder AIL and **13%** over Independent-Encoder AIL; at 1M steps, the averaged performance gain over Shared-Encoder AIL becomes **11%** and over Independent-Encoder AIL is **131%**. Hence, we think the performance gain is non-trivial.
>
> Also, the performance gain is even more obvious in more complex environments like Quadruped-Run task, at 1M frames, the performance gain of PatchAIL over Shared-Encoder AIL is **26%** and the gain over Independent-Encoder AIL is **19%**. The performance gain should be considered significant since PatchAIL has already achieved near-optimal (mean score of 423) compared with the expert (mean score of 547).
>
> Moreover, from Fig 4 we can see that on Cartpole-Swing, Finger-Spin, Walker-Walk, and Quadruped-Run, PatchAIL and its variants show better efficiency with no confidence interval overlap. And on Finger-Spin, Walker-Stand, and Quadruped-Run, the convergence performance of PatchAIL are significantly more stable and higher than the baselines.
>
> Besides, we have included results utilizing action information with annealing BC using only one expert trajectory in Appendix B.3 and Figure 9, which reveals that **PatchAIL can achieve the best learning efficiency while remaining stable performance** (by converging to expert performance for more than **4$\times$ speedup** on Cartpole-Swingup and **2$\times$ speedup** on Finger-Spin).
>
> We hope our explanation and the "good empirical performance" comments from all other reviewers can help address your concerns about the performance gain. And we have added these statistics above into our revised paper to emphasize our performance gain.
> ***
> **Q2**: The other claimed advantage of the method is visual interpretability. However, according to Figure 5, both Shared-Encoder as will as the proposed method have their attention maps focusing on the agent.
>
> **A2**: The interpretability of PatchAIL comes from not only the fact that PatchAIL captures better and clearer shapes of the robot, which is significant across all environments (Figure 5); but also that the patch rewards can explain where is good or bad due to the evaluation of local patches (Figure 1).
>
> Moreover, the interpretability of attention maps in Figure 5 needs to be more informative because they only show the activations of neural networks. Therefore, it is more important to give an interpretation of "*which part of the image makes the discriminator judge whether it is expert-level or low-level.*" **Only with the patch rewards mechanism** can we explain which parts of the image are good or bad compared with the expert, as shown in Figure 1, where the patch rewards align with the movement of key elements.
>
> Finally, we emphasize that all methods can draw attention to maps as in Figure 5, but only PatchAIL enables us to visualize the rewards on different local parts of images like Figure 1. We recommend the reviewer check out the video of our demo page https://sites.google.com/view/patchail/.
>
> ***
>
> We hope the above responses address your primary concerns and make you reconsider your evaluation score. If you have any further questions, please let us know.

---

> > ### Comment · Reviewer_AJEj · 2022-11-30
> > **Thank you for your responses**
> >
> > Dear authors,
> > Thank you for your responses.
> > Regarding Q1, thank you for highlighting the performance gains both in the response and in the manuscript. I do want to point out that, compared with Shared-Encoder AIL which has the best asymptotic performance among baselines, reading from Figure 4, it seems that the average performance gain mostly comes from the Finger Spin environment. Therefore it is not convincing that the proposed method has an overall benefit generalizable across environments. With that said, I agree that the method helps with sample efficiency.
> > Regarding Q2, first of all, thank you for sharing the videos on the demo page. However, it is possible that the highlights in the visualization are strongly correlated with moving pixels. Therefore, it is possible that the patch reward helps identify which pixels are changing, instead of which pixels are contributing positively or negatively to the rewards. Compared to the baselines, though, the proposed method does provide more useful signals.
> > I've increased my score accordingly.

---

> > > ### Author Response · Authors · 2022-12-02
> > > **Thanks for your response and we sincerely ask for further checking**
> > >
> > > We first thank you for your response and the positive comments on our contribution for visual explanation. However, regarding to our asymptotic performance, we would like to sincerely ask you for further checking the figure 4 It is obvious that Shared-Encoder AIL behaves worse than ours on tasks like Finger-spin, Hopper-stand and Walker-Stand and Quadruped-Run. On the other environments, Shared-Encoder AIL only achieves competitive results compared to PatchAIL methods.

---

### Official Review · Reviewer_nR86 · 2022-10-31

**Confidence:** 4
**Correctness:** 3
**Technical Novelty And Significance:** 3
**Empirical Novelty And Significance:** 3
**Recommendation:** 8

**Clarity, Quality, Novelty And Reproducibility:**

The paper is overall well-written. The authors do not include their code, so it's hard to verify the reproducibility of the method. I hope the authors can release their code publicly after acceptance since visual adversarial imitation learning is quite difficult and many researchers want to find a good baseline.

**Strength And Weaknesses:**

Strengths:
* This paper investigates a practical setting. Visual adversarial imitation learning is quite difficult and few works can solve this problem.
* The idea is novel. Though dividing images into patches is nothing new, it is a good try to use it in imitation learning.
* Good and convincing empirical results.

Weaknesses:
* In 5.3, the authors claim that using 'mean' is the one-of-the-best choice among aggregation functions while 'max' tends to fail in both domains. But in my opinion, once the authors use 'mean' as the aggregation function, it seems that there is no difference between using the patch discriminator and a vanilla discriminator. Also, in Figure 6(a), the curve clearly shows that using 'mean' as an aggregation function results in the worst performance. This seems contradictory to what the authors describe in the paper.
* The effect of patch regularization is not clear, as shown in Figure 4.
* Would you mind providing some intuitive analysis about why a too-large or too-small patch could hurt the performance? And In my opinion, the best patch size for different agents should be various due to their different body sizes in the image. Can the authors elaborate on this point more?
* Do PatchAIL have to use (s,s') as the input of the discriminator? I think this adds a strong constraint on the used expert demonstrations. I believe there are some works about adversarial imitation learning with observations only use a single state as the discriminator input. What if we use a single (s) or (s,a) as the discriminator input?
* Experiments can be conducted on other visual imitation learning tasks such as Atari Games.

**Summary Of The Paper:**

This paper investigates the problem of visual adversarial imitation learning. Based on Generative Adversarial Imitation Learning (GAIL), the authors re-formulate the discriminator and make it focus on the local regions of the image. Specifically, the visual demonstrations are divided into patches, the discriminator can predict reward for each patch separately. Empirical results show the performance of PatchAIL can outperform the compared methods in several DMC tasks.

**Summary Of The Review:**

Overall, the paper is well-written and easy to follow. The idea is novel and reasonable. Empirical results are good and convincing. However, there are still some concerns about the experiment, as listed in the weaknesses part. As a result, I provide my initial score as *borderline accept*.

---

> ### Author Response · Authors · 2022-11-13
> **Author Reply to Reviewer nR86**
>
> Thanks for your helpful feedback on our work.
> ***
> **Q1**: 'In 5.3, the authors claim that using 'mean' is the one-of-the-best choice among aggregation functions while 'max' tends to fail in both domains. But in my opinion, once the authors use 'mean' as the aggregation function, it seems that there is no difference between using the patch discriminator and a vanilla discriminator. Also, in Figure 6(a), the curve clearly shows that using 'mean' as an aggregation function results in the worst performance. This seems contradictory to what the authors describe in the paper.'
>
> **A1**: We apologize for the labels being mistyped in Figure 6, which has been corrected in the revision. In the revised version of Figure 6, it can be seen that the mean is one-of-the-best choices among these aggregation functions. We argue that using 'mean' is not the same as a vanilla scalar discriminator for the following reasons:
>
>  1)  the training of the patch discriminator requires evaluating the expertise of different local parts of images;
>
>  2)  due to the patch rewards, the aggregation can provide a more precise evaluation of the expertise of the whole image for the agent to train with.
>
> ***
> **Q2**: Would you mind providing some intuitive analysis about why a too-large or too-small patch could hurt the performance? And In my opinion, the best patch size for different agents should be various due to their different body sizes in the image. Can the authors elaborate on this point more?
>
> **A2**: The different patch sizes affect each patch's receptive field. When the patch size is too large (the extreme case is the pixel case), the receptive field is relatively small, and there is not enough local information used for evaluating expertise. In contrast, a small patch size often relates to a relatively large receptive field, resulting in difficult learning because judging upon a large receptive field is similar to independent-encoder AIL which evaluates the whole image directly.
> ***
> **Q3**: Do PatchAIL have to use (s,s') as the input of the discriminator? I think this adds a strong constraint on the used expert demonstrations. I believe there are some works about adversarial imitation learning with observations only use a single state as the discriminator input. What if we use a single (s) or (s,a) as the discriminator input?
>
> **A3**: In Appendix B.1 and Figure 7, we have included ablation studies about using $s$ or $(s,s')$. Our observations are that this choice makes nearly no difference for shared-encoder AIL, as it learns from latent samples. On the contrary, utilizing observation pairs $(s, s′)$ shows better performance for PatchAIL (especially on Cartpole-Swingup, Finger-Spin, and Hopper-Stand).
> Actually, using $(s,s')$ instead of $s$ is not a strong constraint for image-based tasks, where the visual observation is always a stack of image sequences (e.g., a stack of 3 frames in DeepMind Control Suite used in this paper), and using an observation pair $(s,s')$ is just stacking two stacks of images into one bigger stack.
> ***
> **Q4**: About the code and reproduction.
>
> **A4**: Sure! Our code is based on an open-source code base and as mentioned in our reproduction statement (Section 8). **In our revision we have uploaded the codes in supplementary files** and we are committed to open-sourcing them upon publication.

---

> > ### Comment · Reviewer_nR86 · 2022-11-26
> > **Rebuttal acknowledgement**
> >
> > I have read the authors' response and authors have well addressed most of my concerns. But I still think it might be better to investigate  the performance of PatchIL in more visual imitation learning domains, such as Atari Games and automonous driving. Overall, it is a good paper, and the strengths clearly overweigh the weaknesses. As a result, I am increasing my score to 8.

---

> > > ### Author Response · Authors · 2022-12-02
> > > **Thanks for your response!**
> > >
> > > Thank your for your advice! It is true that we are investigating the performance of PatchAIL in difference domains such as Atari games, and we believe it will included in our open source code for comparison in future works.

---

### Official Review · Reviewer_52Vk · 2022-11-02

**Confidence:** 4
**Correctness:** 3
**Technical Novelty And Significance:** 4
**Empirical Novelty And Significance:** 4
**Recommendation:** 8

**Clarity, Quality, Novelty And Reproducibility:**

### Clarity
The contribution and main idea of this work is clearly stated, and maths are easy to follow.

### Quality
The paper shows good empirical performance together with visualization of learned discriminator’s outputs. The contribution of this work is purely on the empirical side, not on theoretical derivation.

### Novelty
The idea of using patch-wise multi-discriminator seems novel and easy to be implemented.

### Reproducibility
Details on hyperparameters and experimental settings are shared, so it seems straightforward to reproduce the results.



**Strength And Weaknesses:**

### Strength
- The idea seems to be easily applicable to any visual imitation learning tasks.
- One can visualize the gap between agent and expert behaviors through images.
### Weaknesses
- Motivation and theoretical derivation on patch regularization is unclear. It seems that patch regularization is effective only for some tasks where algorithms w/o patch regularization is unstable.
- Analysis on how the number of expert demonstrations affects the performance is missing, e.g., sample efficiency w.r.t. the number of expert demonstrations, whether the patch reward can be effectively trained for small or large numbers of expert demonstrations, etc.


**Summary Of The Paper:**

Visual imitation learning algorithm using patch rewards is proposed. For patch rewards, input images from agent and expert are decomposed into small patches and classified by using multiple patch-wise discriminator. The output values of patch-wise discriminators are postprocessed (by $h$) and aggregated (by $Aggr$, e.g., $\mathrm{mean}$, $\mathrm{median}$, $\mathrm{min}$) to generate a scalar reward that is suitable for the RL inner loop of adversarial imitation learning. Authors also introduce patch regularizers that aim to maintain patch-wise information disappearing through the aforementioned reward aggregation. The proposed algorithm shows its effectiveness in DeepMind Control Suite benchmarks and some amount of visual explanability through learned patched rewards.

**Summary Of The Review:**

Overall, I believe this is a good submission and far above the acceptance boundary. I only have a few questions below:
- The approximation from Eq (6) to Eq (7) is unclear to me, especially how $\mathrm{min}$ can be converted into $\mathbb{E}$ in the bracket of Eq (7).
- Adding empirical analysis w.r.t. the number of expert demonstrations will be interesting. I think that may affect Eq (7) in the sense that the expectation in Eq (7) and its approximation can be affected by the number of expert demonstrations.
- In Figure 6, (a), it seems that $\mathrm{mean}$ doesn’t seem to work well, whereas $\mathrm{max}$ works well. This is different from what authors state in Section 5.3. I guess labels in Figure 6 are simply mistyped.

---

> ### Author Response · Authors · 2022-11-13
> **Author Reply to Reviewer 52Vk**
>
> Thanks for your positive feedback on our work. We respond to individual points as follows.
> ***
> **Q1**: 'Analysis on how the number of expert demonstrations affects the performance is missing' 'Adding empirical analysis w.r.t. the number of expert demonstrations will be interesting. I think that may affect Eq (7) in the sense that the expectation in Eq (7) and its approximation can be affected by the number of expert demonstrations.'
>
> **A1**: In Appendix B.2 and Figure 8, we show the results with fewer demonstrations (*5 trajectories*), which reveals consistent conclusions as in Section 5.1. Besides, we have also included results utilizing action information with annealing BC using only *1 expert trajectory* in Appendix B.3 and Figure 9, showing that PatchAIL methods can achieve the best learning efficiency while remaining stable.
> ***
> **Q2**: The approximation from Eq (6) to Eq (7) is unclear to me, especially how $\texttt{min}$ can be converted into $\mathbb{E}$  in the bracket of Eq (7).
>
> **A2**: We apologize for the inaccurate statement. In our revision, we first change the "*approximation*" statement to "*As a replacement, we can choose to compute a simplified similarity ... instead*".
> Then, we supplement an experiment in the Appendix to compare the effect of equation (6) with equation (7) in Appendix B.5 and draw conclusions that:
>
>  *  Eq (7) and Eq (6) achieve similar similarity trends and Eq (6) obtains larger similarity values on average;
>
>  *  Eq (6) can be more stable when there are fewer expert data and can converge to better final results.
>
> We plan to test more experiments using Eq (6) in the next few days (as of Nov 12, 2022) and will update our results once we obtain new results.
> ***
> **Q3**: In Figure 6, (a), it seems that $\texttt{mean}$ doesn’t seem to work well, whereas works well. This is different from what the authors state in Section 5.3. I guess the labels in Figure 6 are simply mistyped.
>
> **A3**: Thanks for pointing it out! We apologize for the labels being mistyped in Figure 6, which has been corrected. In the revised version of Figure 6, it can be seen that the mean is one-of-the-best choices among these aggregation functions.

---

### Official Review · Reviewer_psvi · 2022-11-03

**Confidence:** 4
**Correctness:** 3
**Technical Novelty And Significance:** 2
**Empirical Novelty And Significance:** 3
**Recommendation:** 6

**Clarity, Quality, Novelty And Reproducibility:**

The paper is clearly written. Though the technical novelty is limited given the algorithm, the empirical noverlty is very high. The authors not only introduce a new algorithm. They show why the performance gained from their proposed algorithm can not be achieved from baseline algorithms through several ablations of the baselines. The work is high quality and provides insight to the community.

**Strength And Weaknesses:**

Strength:
- The idea of training a  discriminator with patches to gather dense rewards straightforward
- The paper is mostly well-written and easy to understand
- The authors perform thorough experiments comparing all relevant baselines to showcase the significance of their idea.
- The authors perform an ablation study in figure 6 (b) on various patches, empirically showing why patch size 39x39 was chosen.
- Additional ablation studies in the appendix show the breadth of empirical analysis.

Weakness:
- Though training a discriminator with patches is easy and intuitive, the best way to aggregate patches into a reward is not apparent. The authors proposed three aggregation metrics: mean, max, min, and median. From this set of metrics mean was chosen as the default aggregation metric. But from the ablation study, figure 6 (a), mean seems to perform significantly worse than the other metrics. It is not apparent why mean was chosen.
- There are 3 variants of PatchAIL: PatchAIL w.o. Reg, PatchAIL-W and PatchAIL-B. It is not apparent which algorithm variant is better and why regularization helps. The authors motivate patch regularization, but the experiment evaluation does not justify the motivation.
- The approximation of equation (6) with equation (7) is unclear.

**Summary Of The Paper:**

This paper describes an adversarial imitation learning algorithm that learns a discriminator with patch pixel rewards rather than directly from images. Recent work in the vision deep learning community have shown that using patches is effective in embedding pixels. The authors argue that patch pixels rewards are more informative than using a shared encoder for the discriminator and critic or directly learning from images.

Comments:
- Is the encoder layer CNN and kernels the same size for the independent-Encoder and Shared-Encoder compared to the patch (obviously ignore the 1x1 kernel at the end)?
- What is \boldmath{R}_{pxp} in equaiton 5?
- Which of the 7 tasks were used for tuning hyperparameters versus for evaluation? Or where all hyperparameters run on all 7 tasks and the best hyperparameter across tasks chosen?

**Summary Of The Review:**

I recommend this paper for marginally above the acceptance threshold. Though the technical novelt is limited, the empirical novelty is not. The authors do a thorough job empirically justifying the decision made in the paper.

---

> ### Author Response · Authors · 2022-11-13
> **Author Reply to Reviewer psvi**
>
> We thank the reviewer for your comments. We hope that we can address your concerns below.
> ***
> **Q1**: 'Is the encoder layer CNN and kernels the same size for the independent-Encoder and Shared-Encoder compared to the patch (obviously ignore the 1x1 kernel at the end)?'
>
> **A1**: We apologize for missing such important information in the paper. The details have been updated in the revision.
> As for the detailed encoder architecture, all methods use the default structure for the policy/critic that works in [1]: a 4-layer CNN with kernels [3$\times$3, 32, 2, 0], [3$\times$3, 32, 1, 0], [3$\times$3, 32, 1, 0], [3$\times$3, 32, 1, 0] (represented as [size, channel, stride, padding]), resulting in a 35$\times$35 feature map and each entry has a receptive field of 15. Note that the encoder-based baselines also utilize such an encoder for the discriminator.
>
> After encoding, the feature map is flattened as a 39,200-length vector, which is then put into a 4-layer MLP actor and a 4-layer critic. For encoder-based baselines, the discriminator also follows the structure in [1] as a 3-layer MLP which outputs a scalar result. For PatchAIL methods, the discriminator is a 4-layer FCN with default kernels [4$\times$4, 32, 2, 1], [4$\times$4, 64, 1, 1], [4$\times$4, 128, 1, 1] and [4$\times$4, 1, 1, 1] (represented as [size, channel, stride, padding]), resulting in 39$\times$39 patches and each has a receptive field of 22.
>
> The above statements are now supplemented in Appendix C.1.
>
> [1] Haldar S, Mathur V, Yarats D, et al. Watch and match: Supercharging imitation with regularized optimal transport . arXiv preprint arXiv:2206.15469, 2022.
> ***
> **Q2**: 'What is \boldmath{R}_{pxp} in equaiton 5?'
>
> **A2**: As shown in the equation, $\textbf{R}_{P \times P}(s, s')$ is the patch reward computed by the output logits of the patch discriminator.
> ***
> **Q3**: Which of the 7 tasks were used for tuning hyperparameters versus for evaluation? Or where all hyperparameters run on all 7 tasks and the best hyperparameter across tasks chosen?
>
> **A3**: The hyperparameters were tuned on Cartpole-Swingup and Finger-Spin, which were then applied to all 7 tasks.
> ***
> **Q4**: It is not apparent which algorithm variant is better. The authors motivate patch regularization, but the experiment evaluation does not justify the motivation. The approximation of equation (6) with equation (7) is unclear.
>
> **A4**: From our main experiments, patch regularization helps stabilize the training in some experiments, such as Finger-Spin, which is also the main task on which we tune our algorithms. However, we find such an unstable problem occurs less in other tasks. Therefore, we revise our statement in the last paragraph of Section 5.1 as "*PatchAIL becomes more stable, which is quite evident in the Finger-Spin domain, yet the main performance gain comes from the patch reward structure given the overall results*".
> As for eq (6) and eq (7), in our revision, we first change the "*approximation*" statement to "*As a replacement, we can choose to compute a simplified similarity ... instead*".
>
> Moreover, we supplement an experiment in the Appendix to compare the effect of equation (6) with equation (7) in Appendix B.5 and draw conclusions that:
>
>   *  Eq (7) and Eq (6) achieve similar similarity trends and Eq (6) obtains larger similarity values on average;
>
>   *  Eq (6) can be more stable when there are fewer expert data and can converge to better final results.
>
> We plan to test more experiments using Eq (6) in the next few days (as of Nov 12, 2022) and will update our results once we obtain new results.

---

> > ### Comment · Reviewer_psvi · 2022-12-07
> > **Rebuttal acknowledgement**
> >
> > Thank you to the authors for addressing my questions and concerns thoroughly. With the authors' changes, the paper's clarity has improved. I maintain my positive view of the paper.

---

### Author Response · Authors · 2022-11-25
**We are looking forward to your response.**

Dear reviewers,

First, thank you again for your valuable comments and suggestions. In the previous responses, we tried our best to address the point of your questions and supplemented more experiments. For example, we revise the mistyped labels in Fig 6 (for Reviewer 52Vk and nR86); we justify the concern about the approximation of equation (6) with equation (7) by revising the statement and supplementing experiments (for Reviewer 52Vk and psvi); we clarify our contribution on both performance improvement and visual interpretability (for reviewer AJEj), along with other particular questions.
We sincerely look forward to your reply to our response. Moreover, we are open to any discussion to improve our paper.

Best,
Authors

---

### Author Response · Authors · 2023-01-09
**Atari experiments are supplemented.**

Dear reviewers,

We have supplemented more results on Atari games, and the corresponding contents are revised in our draft.

The authors

---

### Decision · Program_Chairs · 2023-01-20

**Decision:**

Accept: poster

**Justification For Why Not Higher Score:**

Reviewers appreciated the simplicity of this method and most found the results to be strong enough for acceptance. At the same time, all reviewers recognized that the contribution is mainly an empirical discovery, and at the technical level there is not much new, especially given the popularity of similar methods in the GAN literature. Therefore, the AC feels this paper should be accepted as a poster.

**Justification For Why Not Lower Score:**

See above

**Metareview: Summary, Strengths And Weaknesses:**

Summary:
This paper proposes to use a patch-based discriminator for adversarial imitation learning. The results show gains in performance, stability, and interpretability.

Strengths:
* Simple
* Effective

Weaknesses:
* Path-based discriminators (PatchGANs) are already popular in other applications
* Numerical gains in some experiments are slight

**Note From Pc:**

if the above contains the word "oral" or "spotlight" please see: "oral" presentation means -> notable-top-5% and "spotlight" means -> notable-top-25%. As stated in our emails, we are disassociating presentation type from AC recommendations